# Predictors for Success and Failure in Transoral Robotic Surgery—A Retrospective Study in the North of the Netherlands

**DOI:** 10.3390/cancers16081458

**Published:** 2024-04-11

**Authors:** Alexandra G. L. Toppenberg, Thomas S. Nijboer, Wisse G. W. J. van der Laan, Jan Wedman, Leonora Q. Schwandt, Robert E. Plaat, Max J. H. Witjes, Inge Wegner, Gyorgy B. Halmos

**Affiliations:** 1Department of Ear Nose Throat Surgery, University Medical Centre Groningen, 9713 GZ Groningen, The Netherlands; a.g.l.toppenberg@umcg.nl (A.G.L.T.); t.s.nijboer@umcg.nl (T.S.N.); w.g.w.j.van.der.laan@umcg.nl (W.G.W.J.v.d.L.); j.wedman@umcg.nl (J.W.); i.wegner@umcg.nl (I.W.); 2Department of Otorhinolaryngology–Head and Neck Surgery, Medical Center Leeuwarden, 8934 AD Leeuwarden, The Netherlands; l.q.schwandt@mcl.nl (L.Q.S.); r.plaat@mcl.nl (R.E.P.); 3Department of Oral Maxillo Facial Surgery, University Medical Centre Groningen, 9713 GZ Groningen, The Netherlands; m.j.h.witjes@umcg.nl

**Keywords:** transoral robotic surgery, carcinoma of unknown primary, oropharyngeal cancer, obstructive sleep apnea, chronic lingual tonsillitis

## Abstract

**Simple Summary:**

Transoral Robotic Surgery (TORS) is increasingly used for various conditions, yet its success and failure remain underexplored. In this multicenter study, we assessed 220 TORS interventions across 211 patients. Success criteria differed by condition, including tumor identification for CUP, resection margins for malignancies, and symptom improvement for benign cases like OSA and chronic lingual tonsillitis. Predictors of success varied, with factors such as patient comorbidity, tumor characteristics, and demographics playing roles. Conversely, failure predictors encompassed postoperative complications and pain, linked to patient health and demographics. These insights into TORS outcomes may aid in patient selection and counseling across different conditions, informing clinical practice and enhancing patient care.

**Abstract:**

Transoral Robotic Surgery (TORS) is utilized for treating various malignancies, such as early-stage oropharyngeal cancer and lymph node metastasis of an unknown primary tumor (CUP), and also benign conditions, like obstructive sleep apnea (OSA) and chronic lingual tonsillitis. However, the success and failure of TORS have not been analyzed to date. In this retrospective observational multicenter cohort study, we evaluated patients treated with TORS using the da Vinci surgical system. Success criteria were defined as identification of the primary tumor for CUP, >2 mm resection margin for malignant conditions, and improvement on respiratory polygraphy and tonsillitis complaints for benign conditions. A total of 220 interventions in 211 patients were included. We identified predictors of success, such as low comorbidity status ACE-27, positive P16 status, and lower age for CUP, and female gender and OSA severity for benign conditions. For other malignancies, no predictors for success were found. Predictors of failure based on postoperative complications included high comorbidity scores (ASA) and anticoagulant use, and for postoperative pain, younger age and female gender were identified. This study provides valuable insights into the outcomes and predictors of success and failure in TORS procedures across various conditions and may also help in patient selection and counseling.

## 1. Introduction

Since 2009, Transoral Robotic Surgery (TORS) has primarily been used in the management of oropharyngeal malignancies [1]. While conventional approaches involve (chemo-)radiotherapy, TORS has emerged as a surgical treatment option for patients with early-stage tumors (I and II) [2]. Notably, it provides a viable option for those with recurrent disease or those ineligible for radiotherapy [3]. Furthermore, TORS is employed for identifying the primary tumor in case of lymph node metastasis of an unknown primary tumor (cancer of unknown primary, CUP). Despite the use of PET-CT, the primary tumor remains unidentified in over 50% of cases, underscoring the indispensability of TORS in enhancing identification, as identification rates of 80% are reported [4,5]. Removing the lymphoid tissue of the base of the tongue (BOT) and tonsils (mucosectomy) has proven to be an essential tool in identifying the primary tumor, especially in HPV-related neck metastases [6].

In addition, TORS is used for benign indications, like obstructive sleep apnea (OSA) and chronic lingual tonsillitis. In OSA, TORS is employed for patients with hypertrophy of lingual tonsils causing obstruction of the airway. In these cases, the BOT is reduced, achieving a wider airway and minimizing collapse [7]. This technique has been proven to achieve a surgical success rate between 48% and 68% [8]. Likewise, in patients suffering from chronic lingual tonsillitis, the BOT is reduced to prevent recurrent infections of the lingual tonsils [9]. In both diseases, TORS is described as a safe and effective treatment method.

Despite the known safety and benefits of TORS in treating various malignant and benign conditions, the absence of research on predictors for success and failure in these indications within a larger cohort needs further exploration.

Therefore, this study aims to describe the firsthand experiences with TORS in the north of the Netherlands, combining data from two centers. The primary focus lies in finding predictors for surgical success, like complete tumor resection, finding the primary tumor in CUP, and reduction in symptoms for benign conditions, and failures, like postoperative complications and pain.

## 2. Materials and Methods

This study followed the guidelines detailed in the Strengthening the Reporting of Observational Studies in Epidemiology (STROBE) framework, as outlined in Appendix A [10].

### 2.1. Study Design

The departments of otorhinolaryngology and head and neck surgery of the University Medical Center Groningen (UMCG), a university medical center, and Medical Center Leeuwarden (MCL), a general hospital, in the Netherlands conducted a retrospective observational multicenter cohort study on patients treated with TORS. 

### 2.2. Setting

Patients who underwent TORS using the Xi or Si da Vinci^®^ surgical system (Intuitive Surgical, Inc., Sunnyvale, CA, USA) between November 2018 and December 2023 at the UMCG and between October 2018 and December 2022 at the MCL were included in this analysis. 

This study has been registered in the UMCG research register (PaNaMa, number 18840) and the Institutional Review Board of the UMCG has assessed the study and judged that no approval is needed in accordance with Dutch Medical Research Law legislation. Prior to agreeing to surgery, patients received comprehensive counseling regarding alternative treatments and postoperative complications and provided informed consent for the procedure. Oncological patients were discussed in the multidisciplinary tumor board and were treated according to the current guidelines.

### 2.3. Inclusion

All patients undergoing TORS were included. Before the first inclusion, the head and neck tumor board of the UMCG and MCL designed a multidisciplinary protocol for TORS usage. The protocol for assessing suitability of TORS was established in accordance with the Dutch national guidelines. Patients were selected for TORS in case a malignancy was found in the oropharynx staged as T1 to T2 and in case of CUP. In case of CUP with a suspected lesion in the BOT, patients received targeted biopsies, and no mucosectomy was performed.

OSA patients were eligible for TORS if OSA was proven via respiratory polygraphy and collapse of the airway at the level of the BOT during drug-induced sleep endoscopy. For chronic lingual tonsillitis, patients were eligible for surgery if quality of life was deteriorated due to recurrent tonsillitis and conservative treatment consisting of treatment with antibiotics and proton pump inhibitors was unsuccessful. 

### 2.4. Variables

Data on age, gender, body mass index, smoking, alcohol consumption, the use of anticoagulants, ACE 27 scores, ASA, postoperative feeding, surgery duration—defined from docking the robotic system until the robot is removed from the surgical field—and specific for malignant conditions, P16 status, previous malignancies, histological margins, and adjuvant treatments, and for benign conditions, OSA severity measured by apnea hypopnea index (AHI), oxygen desaturation index (ODI), OSA treatment, and, in case previous objective measures were unavailable, subjective complaint reduction were retrospectively collected from the electronic patient files.

### 2.5. Outcomes

Success outcomes were defined separately for malignant cases, categorized into CUP and other malignancies, as well as for benign cases. Conversely, failure outcomes were considered collectively for all cases, as they pertained to the surgical intervention rather than the underlying disease.

### 2.6. Success Definitions for Malignant Cases

Success was defined as follows: (i) identification of the primary tumor in CUP, as mucosectomies are performed as a diagnostic procedure; (ii) acquire a tumor-free resection margin of more than 2 mm for other malignancies.

### 2.7. Success Definitions for Benign Cases

Success was defined as follows: reduction in symptoms, objective and subjective, for benign conditions. Reduction in symptoms was defined for OSA following Sher’s criteria: AHI reduction of 50% combined with an AHI below 20, compared to baseline [11]. Additionally, a reduction in the ODI 4% of 25% 3 months after surgery was defined as success. For chronic lingual tonsillitis, reduction in the visual analog scores (VASs) of tonsillitis complaints or no complaints after surgery was defined as success. In case no objective data as previously described was present, success was defined to be complaint reduction as described in the notes of the first outpatient clinic visit after TORS. 

### 2.8. Failure Definitions for All Cases

Failure was defined as postoperative complications using the Clavien–Dindo classification (>1) [12] and postoperative pain scores one day after surgery of three or higher on a scale from 0 to 10, with 0 indicating no pain and 10 indicating the worst imaginable pain. 

### 2.9. Statistical Methods

All statistical analyses were performed using an IBM SPSS Statistics Processor (version 28.0.1.0). Following assessment of withdrawals and missing data, the suitability of subjects for analysis was determined, and a description of the study population at baseline was provided. 

Baseline characteristics were described using means with standard deviations (SDs) or medians with interquartile ranges (IQRs), depending on the data distribution. Categorical variables were described in terms of the number and percentages of patients.

The relationship between potential predictors and dichotomous outcomes was examined using univariable binary logistic regression analysis. The outcomes were dichotomous and defined as follows: postoperative complications defined by the Clavien–Dindo score [12] were divided into 0–1 or above 1, postoperative pain scores below 3 or above 3, smoking as current smokers and previous smokers or non-smokers, and previous malignancies as present or not present. Furthermore, alcohol consumption was defined as current usage and previous usage or never. P16 status was defined as positive or negative. AHI was categorized in the following way: 5–15, which defines mild OSA; 15–30, moderate OSA; and >30, severe OSA. To perform dichotomous analysis, groups were defined as AHI < 15 and >15. VAS fatigue and snoring were defined as <8 or >8. Success for malignant indication was defined when resection margin was 2 or more millimeters. For benign indications when Sher’s criteria were met or, in case these were not available, subjective complaint reduction. For benign conditions, analyses were performed on a combination of objective and subjective data or solely objective data.

The sample size was insufficient to perform multivariable analysis. A *p*-value of <0.05 was considered statistically significant.

## 3. Results

### 3.1. Patient Characteristics

A total of 211 patients underwent 220 TORS procedures in UMCG (*n* = 100) and MCL (*n* = 111) and were included for analysis. Patients had different indications for TORS surgery grossly divided into malignant cases (*n* = 84) and benign cases (*n* = 127). The malignant cases were further divided into CUP, BOT carcinoma, and other malignancies. For predictors for success analysis, the BOT carcinomas and other malignancies were combined. The benign cases were divided into OSA, chronic lingual tonsillitis, and other benign pathologies. The mean age of all patients was 54.0 years (SD 15.3), with a range of 18 to 86 years. The cohort comprised 97 females (46%) and 114 males (54%). Regarding clinical characteristics, patients with a malignant indication for TORS scored a median of one and two on the ACE-27 and ASA, respectively. In comparison, patients with a benign indication for TORS scored a median of zero and two on the ACE-27 and ASA, respectively. Most patients in the malignant group were previous smokers (46.4%) and currently using alcohol (60.7%). In the benign group, most patients never smoked (37.8%) and never used alcohol (48.0%). Anticoagulation use was higher in the malignant group (32.3%) compared to the benign group (5.5%). For a comprehensive overview of the patients’ characteristics, see Table 1. 

### 3.2. Surgical Characteristics

Of the 220 surgeries performed, in total, 101 were performed in the UMCG and 119 in the MCL. A total of nine patients underwent more than one TORS procedure. In the malignant group, five patients underwent TORS more than once, and for the benign group, four patients. Most patients underwent a BOT resection. The duration of surgery was on average 62.7 min (SD 31.0). Most TORS procedures had no postoperative complications (67.7%). Postoperative feeding was usually normal (94.5%). Hospitalization duration was on average 3.1 days (SD 2.2). The median postoperative pain score was 3.0 (IQR 2.0–4.0). For more detailed surgical characteristics in the different groups, see Table 2. 

### 3.3. Tumor Characteristics

Of the malignant cases, most tumors were histologically proven to be squamous cell carcinoma. Of the CUP, 45% had a positive P16 status. In the BOT carcinoma group, 45.8% of the tumors had a negative P16 status. This was similar in the group with other malignancies (44.0%). Mean histological margins in the group of other malignancies were the highest (2.7 mm, SD 2.5); in the CUP group, when the primary tumor was found, the histological margin was on average 2 mm (SD 1.6). The histological margins were the smallest in the BOT carcinoma group (1.8 mm). In the BOT carcinoma patients, most patients had a history of malignancy (70.8%). In the CUP group, the most prevalent adjuvant treatment was neck dissection and radiotherapy. In the BOT carcinoma group, nine patients had postoperative radiotherapy (PORT), six underwent a second TORS, and nine were in oncological follow-up. For a detailed understanding of the tumor characteristics, we refer to Table 3.

### 3.4. Benign Disease Characteristics

Pre-operative respiratory polygraph measurements in OSA patients, including AHI and ODI 4%, demonstrated median values of 19.2 and 20.6, respectively. Postoperatively, this was reduced to 10.6 and 12.1, respectively. The median pre-operative VAS tonsillitis score in the chronic lingual tonsillitis group was eight, and after TORS, it decreased to two. For a more detailed description, see Table 4. 

### 3.5. Predictors for Success in Malignant Cases

The average follow-up time for all groups was 22 months (7.5–37.5). The survival status of most patients (64.0%) was alive without disease, as shown in Table 3. For identification of the primary tumor in CUP using TORS, the results are presented in Table 5. Significant predictors for successful TORS for CUP were ACE-27 (OR 0.147 (95% CI: 0.036–0.595)), positive P16 status (OR 11.67 (95% CI: 41.86–73.07)), and age (OR 0.903 (95% CI: 0.826–0.987)). The other examined predictors were not statistically significant. 

For the other malignant tumors, success was defined as a resection margin of >2 mm, the analysis results of which are presented in Table 6. No significant predictors for treatment success were identified.

### 3.6. Predictors for Success in Benign Cases

For benign conditions, success was defined as either a combination of objective complaint reduction when Sher’s criteria are met and subjective complaint reduction or objective complaint reduction alone. The results are presented in Table 7 and Table 8, respectively. No predictors for success were identified if success outcomes were combined. However, for objective complaint reduction, gender (OR 6.118 (95% CI: 1.508–24.826)) and AHI (OR 6.187 (95% CI: 1.198–31.967) and OR 5.850 (95% CI: 1.222–27.994) for 5–15 vs. 15–30 and <15 vs. >15, respectively) were found to influence success significantly. All other analyzed variables did not influence success significantly.

### 3.7. Predictors for Failure

When assessing predictors for postoperative complications, ASA scores and anticoagulation use were significantly correlated with an increased risk. Dichotomic ASA scores (1–2 vs. 3–5) showed an OR of 3.382 (95% CI: 1.386–8.252). Anticoagulation use showed an OR of 2.982 (95% CI: 1.167–7.622). All other variables did not significantly correlate with the development of any complications. For an overview of all tested variables, see Table 9.

For postoperative pain scores, chronic lingual tonsillitis showed an increased OR (2.441 (95% CI: 1.049–5.680), *p*-value of 0.038) compared to CUP. Furthermore, female gender and younger age predicted increased pain scores significantly, with an OR of 1.802 (95% CI: 1.034–3.140) and 0.978 (95% CI: 0.960–0.996), respectively. Other variables did not significantly predict increased pain scores (Table 10).

## 4. Discussion

This is the first paper to describe the predictors of success and failure in TORS for malignant and benign indications in a large series of over 200 cases. In this cohort, we analyzed predictors for success and failure of TORS for both malignant and benign indications.

### 4.1. Predictors for Success in Malignant Cases

For cancer of unknown primary, we have found lower ACE27, positive P16 status, and lower age to significantly predict identification of the primary tumor. It is noteworthy that high ACE27 scores have been described to have a worse prognosis for patients with CUP [13], as well as in cases of surgically treated squamous cell carcinoma of the head and neck [14], for those treated with radiotherapy [15], and within a large cohort encompassing all head and neck cancer [16]. In line with these findings, which have been described in the aforementioned studies regardless of TORS, we have found that lower ACE27 scores increase the odds of identifying the primary tumor in CUP patients. P16 status is generally known to be associated with better prognostic outcomes in CUP patients [17], as well as in oropharyngeal squamous cell carcinoma [18]. Although negative P16 status is known to be predictive for treatment failure, failure is differently defined in other studies, referring to oncological outcome, i.e., survival, whereas we describe failure to be postoperative complications or pain [19]. Important to note is that in our cohort, the number of P16-positive cases is only 40%, compared to 91.6% in the literature. This is due to the indication for TORS in our centers, where all PET-negative tumors are eligible for TORS, as it is used as a diagnostic procedure, not an interventional procedure. Furthermore, equal to treatment with (chemo-)radiotherapy, TORS alone is known to be associated with excellent oncological and functional outcomes [20,21]. Combining this with our finding that P16 status is a predictor for identification of the primary tumor, possibly diagnostic mucosectomy, using TORS in patients with CUP should be limited to only P16-positive cases. Finally, in line with our study, lower age has previously been described in oncological studies to be associated with better overall survival [22,23,24,25], similar to identification of the primary tumor [26]. Nevertheless, it is important to consider that our analysis solely focused on calendar age, thus overlooking considerations of frailty that could influence overall survival. In our study, we identified a relatively low number of primary tumors using TORS. This is due to a highly detailed pre-operative workup, consisting of PET-CT and thorough endoscopy including narrow-band imaging.

### 4.2. Predictors for Success in Benign Cases

The absence of predictors for success in benign conditions is in contrast to the expectations, drawn from our results of malignant conditions. However, it should be noted that patients with benign indications typically exhibit better physical status, as evidenced by lower ASA and ACE-27 scores, along with younger age profiles. A possible explanation is that the benign group is more heterogeneous than the malignant group on success outcomes. Notably, objective success measured through respiratory polygraphy reveals that AHI > 15 emerges as a predictor for success. This correlation could be explained by the severity of OSA, where TORS interventions yield a more pronounced effect, as supported by a systematic review indicating higher success rates in patients with severe OSA [8].

### 4.3. Predictors for Failure

Our analysis showed that TORS is a safe procedure as we reported clinically relevant (grade II or more severe) complications in 9% (grade II = 7, grade III = 10, grade IV = 3) of the cases. In total, 12% of the cases were classified as Clavien–Dindo grade I complications, of which most were postoperative hemorrhage. The results confirm our hypothesis that the use of anticoagulants increases the chances of complications including postoperative hemorrhage. As is known, we found that anticoagulant usage, despite appropriate bridging during surgery, increases chances for developing postoperative complications. Therefore, it remains important to follow the standardized protocols with respect to anticoagulation use during surgery and to monitor patients with increased risks more severely. Additionally, patients with higher ASA scores were associated with increased complication rates. Previously, ASA scores have also been identified to predict major complications in TORS, similarly to our findings [27]. Interestingly, ACE-27 is also known to be a predictor for postoperative complications [14]. In this cohort, we did not find the ACE-27 score to be predictive for developing complications. This may be due to the fact that for the outcome postoperative complications, the entire cohort has been analyzed as opposed to only malignant cases. Generally, patients undergoing TORS for benign indications are known to have a lower number of comorbidities and may therefore have skewed the analysis for comorbidities to be indicative for developing complications.

Regarding postoperative increased pain scores, we have found that gender and age were statistically significant predictors. Females are known to have increased postoperative pain scores, confirming our findings [28]. Also, age is known to influence postoperative pain scores, as confirmed by an extensive pain study, where an inverse correlation between age and pain scores was found [29]. A striking finding is that patients with chronic lingual tonsillitis had a 2.44 times higher chance to experience more postoperative pain, compared to patients with CUP. Of all indications, CUP is expected to give the highest postoperative pain scores, as in these patients, lymphoid tissue of the whole BOT was removed. When comparing to the chronic lingual tonsillitis and OSA or BOT resections, the biggest difference is the postoperative defect in the mucosa. Another large difference between the CUP patient and chronic lingual tonsillitis patients is their nature of indication (malignant vs. benign). This most likely causes the difference in postoperative pain scores, as patients with a malignancy are generally more satisfied with their respective surgical outcome as opposed to chronic lingual tonsillitis patients, for which the indication for TORS is less strict.

### 4.4. Strengths and Limitations

A major strength of our study is the extensive sample size, encompassing 220 cases of TORS procedures conducted across the north of the Netherlands, representing a significant portion of TORS centers in the country. This enhances the reliability of the findings. Additionally, combining data from an academic center and a non-academic center improves the representativeness and reliability of the findings. Furthermore, the analysis was thorough, utilizing validated scores such as the Clavien–Dindo classification system, ASA, ACE-27, and respiratory polygraphy.

Nevertheless, inherent limitations that are common in retrospective studies, such as potential for missing data and selection biases, are acknowledged. Due to the heterogeneous group of 211 patients, with different indications for TORS, complexities in interpretation were introduced, potentially compromising generalizability.

Despite these constraints, the findings contribute valuable insights into TORS outcomes in academic and non-academic settings.

## 5. Conclusions

By analyzing possible factors contributing to either success or failure of TORS, this study might help increase success and decrease failure chances in the future. The results may also help in patient selection and consulting.

## Figures and Tables

**Table 1 cancers-16-01458-t001:** Patient characteristics.

	CUP *n* = 40	BOT Carcinoma *n* = 20	Other Malignancies *n* = 24	Total Malignant Cases *n* = 84	OSA *n* = 48	Chronic Lingual Tonsillitis *n* = 67	Other Benign Pathologies *n* = 12	Total Benign Cases *n* = 127
Center, *n* (%)								
UMCG	33 (82.5)	14 (70.0)	12 (50.0)	59 (70.2)	1 (2.1)	31 (46.3)	9 (75.0)	41 (32.3)
MCL	7 (17.5)	6 (30.0)	12 (50.0)	25 (29.8)	47 (97.9)	36 (53.7)	3 (25.0)	86 (67.7)
Age, mean (SD)	63 (9.0)	64.2 (8.3)	67.3 (11.7)	64.6 (9.7)	47.0 (11.2)	48.9	58.3 (12.3)	47.0 (14.3)
Gender, *n* (%)								
Female	14 (35.0)	6 (30.0)	8 (33.3)	28 (33.3)	9 (18.8)	52 (77.6)	8 (66.7)	69 (54.3)
Male	26 (65.0)	14 (70.0)	16 (66.7)	56 (66.7)	39 (81.3)	15 (22.4)	4 (33.3)	58 (45.7)
Comorbidities (ACE-27), median (IQR)	1 (0–3)	2 (0–3)	1 (0–3)	1 (0–2)	0 (0–3)	0 (0–2)	1 (0–2)	0 (0–1)
ASA, median (IQR)	2 (1–4)	2 (1–3)	2.5 (1–4)	2 (2–3)	2 (1–3)	2 (1–3)	2 (1–3)	2 (2–2)
Smoking, *n* (%)								
Current	13 (32.5)	5 (25.0)	10 (41.7)	28 (33.3)	13 (27.1)	16 (23.9)	1 (8.3)	30 (23.6)
Previous	20 (50.0)	9 (45.0)	10 (41.7)	39 (46.4)	8 (16.7)	27 (40.3)	3 (25.0)	38 (29.9)
Never	7 (17.5)	3 (15.0)	2 (8.3)	12 (14.3)	25 (52.1)	17 (25.4)	6 (50.0)	48 (37.8)
Unknown	-	3 (15.0)	2 (8.3)	5 (6.0)	2 (4.2)	7 (10.4)	2 (16.7)	11 (8.7)
Alcohol, *n* (%)								
Current	25 (62.5)	13 (65.0)	13 (54.2)	51 (60.7)	15 (31.3)	15 (22.4)	4 (33.3)	34 (26.8)
Previous	4 (10.0)	1 (5.0)	5 (20.8)	10 (11.9)	7 (14.6)	8 (11.9)	2 (16.7)	17 (13.4)
Never	9 (22.5)	3 (15.0)	4 (16.7)	16 (19.1)	25 (52.1)	34 (50.8)	2 (16.7)	61 (48.0)
Unknown	2 (5.0)	3 (15.0)	2 (8.3)	7(8.3)	1 (2.0)	10 (14.9)	(33.3)	15 (11.8)
Anticoagulation, *n* (%)								
Platelets aggregation inhibitors	9 (22.5)	5 (25.0)	5 (20.8)	19 (22.6)	1 (2.1)	2 (3.0)	2 (16.7)	5 (3.9)
Coumarins	1 (2.5)	1 (5.0)	1 (4.2)	3 (3.6)	-	-	-	-
Heparins	-	-	-	-	-	-	1 (8.3)	1 (0.8)
DOACs	2 (5)	-	3 (12.5)	5 (6.0)	-	1 (1.5)	-	1 (0.8)
None	28 (70.0)	14 (70.0)	15 (62.5)	57 (67.8)	47 (97.9)	64 (95.5)	9 (75.0)	120 (94.5)

Notes: CUP: carcinoma of unknown primary; BOT: base of tongue; OSA: obstructive sleep apnea; SD: standard deviation; IQR: interquartile range; DOACs: direct oral anticoagulants.

**Table 2 cancers-16-01458-t002:** Surgical characteristics.

	CUP *n* = 40	BOT Carcinoma *n* = 24	Other Malignancies *n* = 25	OSA *n* = 51	Chronic Lingual Tonsillitis *n* = 67	Other Benign Pathologies *n* = 13
Center, *n* (%)						
UMCG	33 (82.5)	17 (70.8)	12 (48.0)	1 (2.0)	28 (41.8)	10 (76.9)
MCL	7 (17.5)	7 (29.2)	13 (52.0)	50 (98.0)	39 (58.2)	3 (23.1)
Type of surgery, *n* (%)						
Tonsillectomy	1 (2.5)	1(4.1)	-	-	-	1 (8.3)
BOT resection	12 (30.0)	23(95.9)	-	31 (60.8)	58 (86.5)	2 (16.7)
Combination	27 (67.5)	-	-	20 (39.2)	5 (7.5)	-
Pharyngectomy	-	-	17(68.0)	-	1 (1.5)	2 (16.7)
Other	-	-	8 (32.0)	-	3 (4.5)	9 (69.3)
Duration surgery in minutes, mean (SD)	60(28.3)	77.7 (51.2)	63.4 (36.4)	72.5 (21.9)	53.3 (24.4)	51.5 (19.7)
Postoperative complications following the Clavien–Dindo classification, *n* (%)						
Grade I	1 (2.5)	-	3 (12.0)	4 (7.8)	6 (9.0)	13 (100)
Grade II	-	2 (8.3)	1 (4.0)	1 (2.0)	3 (4.5)	-
Grade III	4 (10.0)	1 (4.2)	2 (8.0)	2 (3.9)	1 (1.5)	-
Grade IV	1 (2.5)	1 (4.2)	-	1 (2.0)	-	-
Grade V	-	-	-	-	-	-
None	34 (85.0)	20 (83.3)	19 (76.0)	43 (84.3)	56 (83.6)	-
Postoperative complications, *n* (%)						
Hemorrhage	1 (2.5)	2 (8.3)	1 (4.0)	6 (11.8)	-	-
Infection	-	2 (8.3)	1 (4.0)	1 (2.0)	1 (1.5)	-
Non-surgical complications	-	-	-	1 (2.0)	1 (1.5)	-
None	34 (85.0)	20 (83.4)	22 (88.0)	39 (76.5)	56 (83.6)	12 (92.3)
Other	-	-	-	4 (7.8)	1 (1.5)	1 (7.7)
Missing	5 (12.5)	-	1 (4.0)	-	8 (11.9)	-
Postoperative feeding, *n* (%)						
Normal	40 (100)	20 (83.3)	18 (72.0)	51 (100.0)	67 (100.0)	12 (92.3)
Tube	-	4 (16.7)	7 (28.0)	-	-	1 (7.7)
Hospitalization duration, mean (SD)	1.9 (1.4)	4.1 (3.9)	3.9 (3.5)	3.2 (1.1)	3.2 (1.3)	2.5 (1.8)
Pain score postoperative day 1, mean (SD)	2.9 (1.8)	3.0 (2.4)	2.4 (1.8)	2.8 (2.0)	3.8 (1.9)	2.9 (1.2)
Dichotomous, *n* (%)						
<3	25 (62.5)	14 (58.3)	19 (76.0)	37 (72.5)	31 (46.3)	8 (61.5)
>3	13 (32.5)	9 (37.5)	6 (24.0)	14 (27.5)	36 (53.7)	4 (30.8)
Missing	2 (5.0)	1 (4.2)	-	-	-	1 (7.7)

Notes: CUP: carcinoma of unknown primary; BOT: base of tongue; OSA: obstructive sleep apnea; SD: standard deviation.

**Table 3 cancers-16-01458-t003:** Tumor characteristics.

	CUP *n* = 40	BOT Carcinoma *n* = 24	Other Malignancies *n* = 25
Histology, *n* (%)			
Squamous cell carcinoma	38 (95.0)	19 (79.2)	20 (80.0)
Other	2 (5.0)	5 (20.8)	5 (20.0)
P16 status, *n* (%)			
Positive	18 (45.0)	9 (37.5)	4 (16.0)
Negative	10 (25.0)	11 (45.8)	11 (44.0)
Unknown	10 (25.0)	3 (12.5)	7 (28.0)
Missing	2 (5.0)	1 (4.2)	3 (12.0)
Histological margin, mean (SD)	2.0 (1.6)	1.8 (2.2)	2.7 (2.5)
History of other malignancies, *n* (%)			
Present None	- 40 (100)	17 (70.8) 7 (29.2)	10 (40.0) 15 (60.0)
Adjuvant treatment, *n* (%)			
Neck dissection	2 (5.0)	-	-
PORT	13 (32.5)	9 (37.5)	5 (20.0)
POCRT	7 (17.5)	-	2 (8.0)
Neck dissection and RT	14 (35.0)	-	1 (4.0)
Other	4 (10.0)	15 (62.5)	17 (68.0)
Follow-up time, median (IQR)	20 (7–36)	22 (5–38)	28 (11–40)
Survival status, *n* (%)			
AWD	2 (5.0)	4 (16.7)	3 (12.0)
AWoD	28 (70.0)	16 (66.7)	14(56.0)
DOOC	3 (7.5)	3 (12.5)	3 (12.0)
DOD	6 (15.0)	1 (4.2)	3 (12.0)

Notes: CUP: carcinoma of unknown primary; BOT: base of tongue; SD: standard deviation; IQR: interquartile range; PORT: postoperative radiotherapy; POCRT: postoperative chemoradiation; RT: radiotherapy; AWD: alive with disease; AWoD: alive without disease; DOOC: death of other causes; DOD: death of disease.

**Table 4 cancers-16-01458-t004:** Benign disease characteristics.

	OSA *n* = 51	Chronic Lingual Tonsillitis *n* = 67	Other Benign Pathologies *n* = 13
AHI, median (IQR)			
Pre-operative	19.2 (11.0–25.9)	13.1 (8.6–16.3)	-
Missing, *n*	1	60	13
Postoperative	10.6 (4.7–22.5)	8.5 (6.2–10.9)	-
Missing, *n*	7	5	13
ODI 4%			
Pre-operative	20.6 (13.5–30.9)	15.9(9.5–71.3)	-
Missing, *n*	3	61	13
Postoperative	12.1(5.3–23.9)	9.6 (5.9–13.0)	-
Missing, *n*	7	62	13
VAS tonsillitis, median (IQR)			
Pre-operative	-	8.0 (7.0–8.3)	-
Missing, *n*	51	33	13
Postoperative	4.0 (4.0–4.0)	2.0 (1.0–4.5)	-
Missing, *n*	50	33	13

Notes: OSA: obstructive sleep apnea; AHI: apnea hypopnea index; ODI 4%: oxygenation desaturation index 4%; VAS: visual analog scale; IQR: interquartile range.

**Table 5 cancers-16-01458-t005:** Univariable analysis for predictors of success in CUP.

Predictor	Significance	Odds Ratio
Gender	0.842	0.875 (0.236–3.241)
Smoking	0.345	2.353 (0.398–13.900)
Alcohol	0.575	0.650 (0.144–2.927)
ACE27	0.007 *	0.147 (0.036–0.595)
Previous malignancies H&N	N.A.	
P16 status	0.09 *	11.667 (1.863–73.066)
Age	0.024 *	0.903 (0.826–0.987)
BMI	0.695	0.979 (0.883–1.087)
Surgery time	0.817	1.003 (0.981–1.025)

Notes: CUP: carcinoma of unknown primary; ACE27: Adult Co-morbidity Evaluation 27; H&N: head and neck; BMI: body mass index. * *p* < 0.05.

**Table 6 cancers-16-01458-t006:** Univariable analysis for predictors of success in malignant conditions based on resection margins.

Predictor	Significance	Odds Ratio
Gender	0.244	0.410 (0.092–1.834)
Alcohol	0.651	0.667 (0.115–3.872)
ACE27	0.537	1.750 (0.296–10.340)
Previous malignancies H&N	0.681	1.310 (0.363–4.728)
P16 status	0.568	0.646 (0.144–2.899)
Age	0.184	0.954 (0.890–1.023)
BMI	0.971	1.002 (0.886–1.134)
Surgery time	0.137	1.011 (0.996–1.026)

Notes: ACE27: Adult Co-morbidity Evaluation 27; H&N: head and neck; BMI: body mass index.

**Table 7 cancers-16-01458-t007:** Univariable analysis for predictors of success in benign conditions.

Predictor	Significance	Odds Ratio
Gender	0.065	2.343 (0.947–5.797)
Smoking	0.641	0.810 (0.334–1.963)
Alcohol	0.552	0.760 (0.308–1.878)
ACE27	0.106	0.482 (0.199–1.168)
Previous malignancies H&N	0.787	0.727 (00072–7.303)
Age	0.797	0.996 (0.966–1.027)
BMI	0.929	0.996 (0.906–1.095)
Surgery time	0.758	1.003 (0.985–1.021)

Notes: ACE27: Adult Co-morbidity Evaluation 27; H&N: head and neck; BMI: body mass index.

**Table 8 cancers-16-01458-t008:** Univariable analysis for predictors of objective success in benign conditions.

Predictor	Significance	Odds Ratio
Gender male vs. female	0.011 *	6.118 (1.508–24.826)
Smoking	0.431	0.611 (0.179–2.081)
Alcohol	0.431	0.611 (0.179–2.081)
ACE27	0.273	0.519 (0.160–1.676)
AHI (5–15, 15–30, >30)	0 vs. 1: 0.030 *	6.187 (1.198–31.967)
	0 vs. 2: 0.270	4.5 (0.310–65.229)
AHI (<15 vs. >15)	0.027 *	5.850 (1.222–27.994)
VAS fatigue < 8 vs. ≥8	0.204	2.571 (0.598–11.059)
VAS snoring < 8 vs ≥8	0.177	3.56 (0.79–1.594)
VAS tonsillitis < 8 vs. ≥8	N.A.	
Age	0.774	0.994 (0.951–1.038)
BMI	0.138	0.894 (0.771–1.037)
Surgery time	0.803	0.997 (0.997–1.020)

Notes: ACE27: Adult Co-morbidity Evaluation 27; AHI: apnea hypopnea index; VAS: visual analog Scale; BMI: body mass index. * *p* < 0.05.

**Table 9 cancers-16-01458-t009:** Univariable analysis for predictors of failure based on postoperative complications in all TORS indications.

Predictor	Significance	Odds Ratio
Surgery type CUP vs. other indications **	CUP vs. 2: 0.644	1.400 (0.337–5.8210)
	CUP vs. 3: 0.952	0.955 (0.207–4.397)
	CUP vs. 4: 0.864	1.114 (0.325–3.812)
	CUP vs. 5: 0.338	0.509 (0.128–2.026)
	CUP vs. 6: N.A.	
	CUP vs. 7: 0.638	0.583 (0.062–5.506)
Gender	0.748	0.869 (0.368–2.051)
Smoking	0.705	0.833 (0.324–2.143)
Alcohol	0.546	1.371 (0.492–3.822)
Postoperative feeding	0.125	2.952 (0.741–11.763)
ASA score	0.007 *	3.382 (1.386–8.252)
ACE27	0.485	1.358 (0.575–3.204)
Anticoagulation use	0.022 *	2.982 (1.167–7.622)
Pain score	0.212	1.757 (0.725–4.261)
Pain medication	0.640	0.689 (0.144–3.288)
Age	0.646	1.007 (0.979–1.035)
BMI	0.257	1.044 (0.969–1.126)
Surgery time	0.208	1.008 (0.996–1.020)
Previous malignancies H&N	0.114	2.267 (0.822–6.254)

Notes: CUP: carcinoma of unknown primary; ASA: American Society of Anesthesiologists; ACE27: Adult Co-morbidity Evaluation 27; BMI: body mass index; H&N: head and neck; * *p* < 0.05; ** 2 = base of tongue carcinoma, 3 = other malignant pathologies, 4 = obstructive sleep apnea (OSA), 5 = benign chronic lingual tonsillitis, 6 = benign OSA and chronic lingual tonsillitis, 7 = benign other.

**Table 10 cancers-16-01458-t010:** Univariable analysis for predictors of failure based on postoperative pain scores in all TORS indications.

Predictor	Significance	Odds Ratio
Surgery type CUP vs. other indications **	CUP vs. 2: 0.698	1.236 (0.423–3.613)
	CUP vs. 3: 0.390	0.607 (0.195–1.892)
	CUP vs. 4: 0.493	0.728 (0.293–1.807)
	CUP vs. 5: 0.038 *	2.441 (1.049–5.680)
	CUP vs. 6: 0.859	1.154 (0.238–5.605)
	CUP vs. 7: 0.955	0.962 (0.243–3.802)
Surgery type malignant vs. benign	0.184	1.472 (0.832–2.603)
Gender	0.038 *	1.802 (1.034–3.140)
Smoking	0.408	1.294 (0.702–2.386)
Alcohol	0.727	0.898 (0.491–1.643)
ASA score	0.415	0.745 (0.368–1.512)
ACE27	0.670	0.887 (0.512–1.538)
Postoperative feeding	0.786	1.178 (0.361–3.842)
Age	0.017 *	0.978 (0.960–0.996)
BMI	0.207	1.036 (0.981–1.094)
Surgery time	0.559	0.997 (0.988–1.006)
Previous malignancies	0.926	1.038 (0.475–2.267)

Notes: CUP: carcinoma of unknown primary; * *p* < 0.05; ** 2 = base of tongue carcinoma, 3 = other malignant pathologies, 4 = obstructive sleep apnea (OSA), 5 = benign chronic lingual tonsillitis, 6 = benign OSA and chronic lingual tonsillitis, 7 = benign other; ASA: American Society of Anesthesiologists; ACE27: Adult Co-morbidity Evaluation 27; BMI: body mass index.

## Data Availability

The original contributions presented in the study are included in the article; further inquiries can be directed to the corresponding author.

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
