# Peer review of "Predictors for Success and Failure in Transoral Robotic Surgery—A Retrospective Study in the North of the Netherlands"

_cancers, 2024, doi:10.3390/cancers16081458_

Round 1
Reviewer 1 Report
Comments and Suggestions for Authors
The main question addressed by the research is objective: predict outcomes in TORS. The main objective is original. The specific gap in the field the paper addresses is how to predict the outcomes. Compared with other published material, they pointed very clear to the problems of this tool ( robotic approach )
Your paper is very nice and it presents us with a real issue of interest (head and neck or tnt surgeons). I recomend it strongly to be published
I have a question concerning the duration of each operation |: the time presented is after the location of the robot system or the real time time after the patient is in o.r.? .
The second pointis a sugestion : I do encourage them to split their job in 2 papers : malignant desease use and benign one. I guess it will be more handfull.
Reviewer 2 Report
Comments and Suggestions for Authors
I read with great interest this manuscript on “Predictors for Success and Failure in Transoral Robotic Surgery – a retrospective study in the North of the Netherlands”. I congratulate the authors for the abundant series and for the meticulous analysis carried out, even if reading the results is a bit tiring due to the heterogeneity of the series (malignant tumors and non-tumor pathologies), in turn divided into 3 categories each.
The result of the analysis for malignant tumors revealed that low comorbidity status ACE-27, positive P16 status and lower age are predictors of success.
These factors are known from an infinite series of previous studies on oncological surgery, and in particular on head and neck surgery, regardless of the use of traditional techniques or TORS.
I therefore think that this fact should be underlined in the discussion.
As regards the use of TORS in neoplastic surgery of the head and neck and, in particular, of the oropharynx, I think that this technique is very useful as an alternative to CT-RT. I think, however, that the pros and cons of each of the two therapies should not be omitted.
In the manuscript, at the lines 250-260 there is the statement: “Furthermore, TORS alone is known to be associated with better oncological and functional outcome compared to patients treated with (chemo)radiotherapy [16]”. This seems to me to be a statement not supported by irrefutable demonstrations. In fact, the same review cited by the authors themselves at number 16 says verbatim: “the evidence in the literature is not conclusive, considering that studies randomizing HPV-positive and HPV-negative patients between primary surgery and radiotherapy are lacking”. And it concludes: “Although TORS is often considered the preferred treatment for T1-T2 N0-N1 HPV+ OPSCC, our systematic review raises several concerns about the validity of this assumption”.
In addition to what was said in the aforementioned review, the results of what, to my knowledge, is the only randomized trial comparing TORS versus CT-RT. The conclusion of this trial was: “Overall survival and Progression free survival were excellent in both groups, and there were no significant differences between treatment arms”. “The percentage of patients receiving total oral diet with no restrictions (on the basis of the functional oral intake scale scores) at 1 year was 100% in the RT arm and 84% in the TORS-ND arm”. The conclusion of the study was: “in contrast to prior retrospective comparisons, RT- and TORS-based approaches were associated with clinically similar QOL”.1
Therefore, I think that the sentence on lines 258-260 should be eliminated or, after it, what was said in the studies reported by me should be added.
1. Nichols AC, Theurer J, Prisman E, Read N, Berthelet E, Tran E, Fung K, de Almeida JR, Bayley A, Goldstein DP, Hier M, Sultanem K, Richardson K, Mlynarek A, Krishnan S, Le H, Yoo J, MacNeil SD, Winquist E, Hammond JA, Venkatesan V, Kuruvilla S, Warner A, Mitchell S, Chen J, Corsten M, Johnson-Obaseki S, Odell M, Parker C, Wehrli B, Kwan K, Palma DA. Randomized Trial of Radiotherapy Versus Transoral Robotic Surgery for Oropharyngeal Squamous Cell Carcinoma: Long-Term Results of the ORATOR Trial. J Clin Oncol. 2022 Mar 10;40(8):866-875. doi: 10.1200/JCO.21.01961. Epub 2022 Jan 7. PMID: 34995124.
Round 2
Reviewer 2 Report
Comments and Suggestions for Authors
I congratulate the authors for accepting my suggestions.
In my opinion, the changes and additions made have improved the manuscript.